# The Differences in the Susceptibility Patterns to Triclabendazole Sulfoxide in Field Isolates of *Fasciola hepatica* Are Associated with Geographic, Seasonal, and Morphometric Variations

**DOI:** 10.3390/pathogens11060625

**Published:** 2022-05-28

**Authors:** Martha V. Fernandez-Baca, Cristian Hoban, Rodrigo A. Ore, Pedro Ortiz, Young-Jun Choi, César Murga-Moreno, Makedonka Mitreva, Miguel M. Cabada

**Affiliations:** 1Sede Cusco, Instituto de Medicina Tropical “Alexander von Humboldt”, Universidad Peruana Cayetano Heredia, Calle Jose Carlos Mariategui J-6, Wanchaq, Cusco 08002, Peru; martha.fernandez.baca.c@upch.pe (M.V.F.-B.); rodrigo.ore@upch.pe (R.A.O.); 2Laboratorio de Inmunología, Facultad de Ciencias Veterinarias, Universidad Nacional de Cajamarca, Avenida Atahualpa 1050, Cajamarca 06001, Peru; cristian.hoban@upch.pe (C.H.); portiz@unc.edu.pe (P.O.); cmurgam15@unc.edu.pe (C.M.-M.); 3Department of Medicine, Division of Infectious Diseases, Washington University School of Medicine, 4523 Clayton Avenue, MSC 8051-0043-15, St. Louis, MO 63110, USA; choi.y@wustl.edu; 4McDonnell Genome Institute, Washington University, 4444 Forest Park Avenue, St. Louis, MO 63108, USA; 5Department of Medicine, Division of Infectious Diseases, School of Medicine, University of Texas Medical Branch, 301 University Boulevard, Galveston, TX 77555, USA

**Keywords:** *Fasciola hepatica*, triclabendazole, resistance, natural infection

## Abstract

Triclabendazole (TCBZ) resistance is an emerging problem in fascioliasis that is not well understood. Studies including small numbers of parasites fail to capture the complexity of susceptibility variations between and within *Fasciola*
*hepatica* populations. As the first step to studying the complex resistant phenotype–genotype associations, we characterized a large sample of adult *F. hepatica* with diverging TCBZ susceptibility. We collected parasites from naturally infected livestock slaughtered in the Cusco and Cajamarca regions of Peru. These parasites were exposed to TCBZ sulfoxide (TCBZ.SO) in vitro to determine their susceptibility. We used a motility score to determine the parasite’s viability. We titrated drug concentrations and times to detect 20% non-viable (susceptible conditions) or 80% non-viable (resistant conditions) parasites. We exposed 3348 fully motile parasites to susceptible (n = 1565) or resistant (n = 1783) conditions. Three hundred and forty-one (21.8%) were classified as susceptible and 462 (25.9%) were classified as resistant. More resistant parasites were found in Cusco than in Cajamarca (*p* < 0.001). Resistant parasites varied by slaughterhouse (*p* < 0.001), month of the year (*p* = 0.008), fluke length (*p* = 0.016), and year of collection (*p* < 0.001). The in vitro susceptibility to TCBZ.SO in wildtype *F. hepatica* was associated with geography, season, and morphometry.

## 1. Introduction

Fascioliasis is a zoonotic trematode infection with wide distribution among humans and livestock. The infection has been reported in 81 developed and developing countries around the world and is believed to affect between 2.4 and 17 million people [1,2,3]. Humans are commonly infected in rural areas of South America, Africa, and Asia [4]. Livestock infections cause decreased fertility and production of meat, milk, and wool associated with significant revenue losses, poverty, and food insecurity [5]. Peru has one of the highest burdens of fascioliasis with a prevalence of up to 70% in livestock and up to 47% among children in some communities of the Andes Mountains [6,7,8,9]. 

TCBZ is the only drug active against juvenile and adult stages of the fluke and is widely used for treatment and control [10,11]. The mechanisms of action are not fully understood, but as with other benzimidazoles, TCBZ specifically binds to beta-tubulin in *F. hepatica* disrupting the microtubules [12]. TCBZ is rapidly metabolized into TCBZ.SO and TCBZ sulfone after absorption in the intestine, with the former being more active and abundant than the latter [12]. Reports of treatment failure and TCBZ resistance have emerged from endemic areas around the world [13,14,15]. Resistance to TCBZ has been reported in the highlands of Peru and documented among humans and livestock. Ortiz et al. in the Northern Peruvian highlands demonstrated a TCBZ efficacy of 25% on a *F. hepatica* isolate obtained from cattle in Cajamarca [16]. Morales et al. reported an efficacy of 70% after two treatment rounds with TCBZ among children with chronic fascioliasis in the Cusco highlands, and 12% of the children failed to show parasitological cure despite repeated treatment courses and high TCBZ doses [13]. The mechanisms of resistance have not been elucidated, but some studies suggested different mechanisms including alterations in drug uptake/efflux, increased metabolism of TCBZ.SO to the less-active sulfone metabolite, and increased glutathione S-transferase detoxification activity [17,18,19,20]. A study by Borgsteede et al. suggested that once resistance to TCBZ emerged in farms, the phenotype persists even after years of stopping its use [21]. Then, the prevention and active surveillance of the spread of TCBZ resistance in endemic areas is paramount to the preservation of effective treatment options and control programs for fascioliasis. 

TCBZ-susceptible and -resistant *F. hepatica* strains collected from livestock have been characterized and maintained in the laboratory for decades [22]. These laboratory adapted *F. hepatica* isolates are used as models for in vitro studies of drug susceptibility [23,24,25,26,27]. Even among the limited number of laboratory parasite clones, significant differences in susceptibility profiles to anthelminthic drugs including TCBZ have been documented. These differences depend on the parasite’s age, drug and drug dose, and geographic location [22]. In addition, differences in parasite fitness assessed by infectious capacity, latent infection periods, and the number of eggs or metacercariae produced have been reported between laboratory isolates with similar resistance patterns [23,28]. These findings suggest that TCBZ resistance is a complex problem that not only leads to treatment failures but may also have broad effects on the parasite’s epidemiology. Thus, understanding how TCBZ resistance arises and spreads in the natural populations of *F. hepatica* is critical for the sustainable control of fascioliasis.

In this study, we report the characterization of a large number of *F. hepatica* parasites collected from naturally infected livestock with diverging TCBZ susceptibility in different endemic areas of Peru. Considering that TCBZ resistance is likely a polygenic/multifactorial trait with substantial variation both within and between populations, studies focusing on a limited number of laboratory or field parasites do not yield the full picture. Bąska et al. demonstrated differences in the expression of immunomodulatory molecules between laboratory isolates of *F. hepatica* and wildtype parasites collected from the field, highlighting the intraspecies variability that may contribute to discrepancies between studies of small sample size [29]. A single nucleotide polymorphism in the P-glycoprotein gene was associated with TCBZ resistance in Ireland, but further studies in Latin America and Australia failed to find the same association [30,31,32]. Our work based on a broad sampling of field isolates identifies parasites at opposite extremes of the TCBZ susceptibility spectrum and is an important first step to a comprehensive evaluation of genotype–phenotype associations in this globally distributed trematode parasite of public health and economic significance.

## 2. Results 

We collected 9613 *F. hepatica* parasites from 430 condemned bovid livers. Parasites from 37 condemned livers were discarded due to decreased viability after 48 h of incubation or bacterial contamination. We selected 3348 *F. hepatica* for exposure to TCBZ.SO, where 98% of these were collected from cattle and 2% were collected from sheep. Most parasites were collected in the Cusco region (61.9%) during 2021 (81.4%) (Table 1). The slaughterhouses where more flukes were collected were Cajamarca (34.9%) in Cajamarca city and Kayra (29.4%) in Cusco city. The mean fluke length was 17.7 mm (± 2.86). Almost equal numbers of flukes were exposed to susceptible conditions (46.7%) or resistant conditions (53.3%). The proportion of flukes exposed to susceptible conditions was lower in 2020 compared to 2021 (32.6% versus 49.9%, *p* < 0.001) and flukes exposed to susceptible conditions were shorter than flukes exposed to resistant conditions (17.3 mm ± 2.66 versus 18 mm ± 3, *p* < 0.001) (Table 1).

Overall, 341 flukes were classified as susceptible, representing 10.2% of all the parasites tested and 21.8% of the parasites exposed to susceptible conditions. The proportion of flukes classified as susceptible varied by slaughterhouse (*p* = 0.024), month of the year (*p* = 0.003), and fluke length (*p* < 0.001) (Table 2). Similarly, 462 flukes were classified as resistant, representing 13.8% of all the parasites tested and 25.9% of the parasites exposed to the resistant conditions. A higher proportion of flukes were classified as resistant in the Cusco region compared to the Cajamarca region (33.8% versus 12.9%, *p* < 0.001). The proportion of flukes classified as resistant also varied by slaughterhouse (*p* < 0.001), month of the year (*p* = 0.008), fluke length (*p* = 0.016), and year of collection (*p* < 0.001) (Table 2). The mean length of resistant flukes was significantly higher than the length of susceptible flukes (18.82 mm (±6.58) versus 16.66 mm (±2.73), *p* < 0.001).

When the parasites collected in 2020 and 2021 were evaluated together, the pattern of variation in the distribution of parasites classified as susceptible or resistant was different depending on the region. In Cusco, the distribution of susceptible *F. hepatica* fluctuated without a clear pattern (*p* = 0.006) while there was a marked increase in the proportion of resistant *F. hepatica* observed between the months of December and July (*p* = 0.001). In contrast, in Cajamarca, a marked peak of susceptible *F. hepatica* was observed between the months of June to September (*p* < 0.001) while the proportion of resistant *F. hepatica* fluctuated throughout the year without a discernible pattern (*p* < 0.001) (Figure 1).

## 3. Discussion

Resistance to TCBZ threatens *F. hepatica* control and treatment efforts in endemic areas in developing and developed countries. The declining TCBZ effectiveness in fascioliasis can impact the epidemiology of the disease in these regions. Characterizing fluke populations by focusing on their TCBZ susceptibility profiles can provide insights into potential effects. This is particularly important in the highlands of Peru where fascioliasis presents a large burden to public health and livestock farming [6,7,8,9]. In the present study, we characterized the susceptibility to TCBZ.SO of *F. hepatica* parasites collected from naturally infected bovines brought for slaughter in representative abattoirs from the Cusco and Cajamarca regions of Peru. TCBZ.SO is the most abundant and active metabolite of TCBZ. The TCBZ metabolite’s activity depends on the tissue concentration of the drug, and this is a function of the drug concentration in the medium and duration of exposure. The experimental conditions in our study considered both variables to account for the slower rates of tissue penetration of TCBZ.SO compared to the parent drug [33]. Thus, prior to our main experiments, we adjusted the TCBZ.SO concentration and exposure time to establish conditions that either inhibited 20% or 80% of the parasites. More than one in four parasites exposed to TCBZ.SO for 24 h were classified as resistant. The distribution of the susceptible and resistant parasites showed variations by location and time of collection. In addition, morphometric differences were associated with susceptibility or resistance to TCBZ.SO. 

A higher proportion of resistant *F. hepatica* was found in the Cusco region compared to the Cajamarca region. Variation in TCBZ use and selective pressure among fluke populations may explain the differences between regions. The livestock industry in the Cajamarca region is among the largest in Peru and ranks third in dairy production [34,35,36]. TCBZ was the main medication used for control and was used 3–4 times a year with only transient effects on infection incidence and intensity. High levels of TCBZ resistance have been widely documented among cattle in Cajamarca [16,37]. However, TCBZ use has been decreasing in the last 6–8 years and is no longer the preferred drug for livestock fascioliasis control in the region, with drugs such as nitroxinil and clorsulon being used more often. Thus, TCBZ selective pressure on livestock fascioliasis has decreased in Cajamarca, possibly in part contributing to identifying a lower proportion of resistant flukes. On the other hand, municipal programs in Cusco’s study areas provide TCBZ to farmers, which is widely used as the drug of choice. High TCBZ usage, deficiencies in dosing, and poor-quality control of veterinary products are likely factors influencing the rapid emergence of resistance in the Cusco area [38].

We observed monthly variations in the proportion of parasites classified as susceptible or resistant, and the patterns were different in Cusco and Cajamarca. There are altitudinal and climatic differences between these two regions that can, in part, explain these differences. Cajamarca in the north of Peru, near the border with Ecuador, is located at a lower altitude and has higher maximal and minimal temperatures than Cusco, located in the south at a higher altitude [39]. Although altitude is associated with a higher prevalence of fascioliasis, temperature and solar radiation are associated with snail populations, infection prevalence, incubation periods, and cercariae production [40,41,42,43]. Resistant and susceptible parasites have also been shown to have different pre-patent periods, production of cercariae per snail, and infectivity capacity, which may give them a transmission advantage depending on the region [23,44,45]. Importantly, patterns of TCBZ use in each region could have also affected the proportions of susceptible and resistant parasites by selective pressure [46,47]. Snail species and competence are other factors not well studied among susceptible and resistant flukes. Further characterization of these field isolates in Cusco and Cajamarca is underway to determine phenotypic and genotypic characteristics associated with virulence. 

The morphometric characteristics of *F. hepatica* have been shown to vary significantly within and between parasite clones [45]. The geographic distribution of the parasites does not seem to play a role in these differences [48]. We used adult *F. hepatica* body length as a representative morphometric characteristic and determined that smaller parasites were more likely to be susceptible to TCBZ.SO than larger parasites. The host species has been associated with fluke size, but it is an unlikely explanation for our findings because 98% of the parasites we tested were collected from cattle [49]. A crowding phenomenon has been proposed in which *F. hepatica* parasites from livers with heavy infections have a smaller size and produce fewer eggs than parasites collected from livers with light infections [50,51]. In our study, we did not perform systematic liver dissections and limited the number of flukes collected to avoid sampling clones from each liver, which precluded the evaluation of crowding as a potential cause of size differences. Other studies have detected size differences between resistant and susceptible *F. hepatica* isolates. McConville et al. using experimental infections in sheep reported that the Sligo TCBZ-resistant isolate was smaller than the Cullompton TCBZ-susceptible isolate [23]. Although McConville et al.’s results contrast with our findings, they may not be directly comparable as we characterized parasites collected in the field, likely representing outcrossing populations. Lastly, the smaller size of the susceptible parasites in our study might reflect the parasite’s age as resistant parasites may live longer in cattle if they were treated with TCBZ at some point.

The scope of this study was limited to the collection of a selected group of parasites obtained mostly from bovines after slaughter in formal abattoirs, which may not directly reflect the spectrum of TCBZ susceptibility at the community level in livestock. However, the proportions of susceptible and resistant parasites in our study provide information about the patterns of TCBZ susceptibility in wildtype *F. hepatica* collected from different endemic areas. The lack of specific cutoffs for TCBZ susceptibility in vitro among wildtype *F. hepatica* forced us to establish definitions empirically. Thus, the reproducibility of our results should be tested in other *F. hepatica* populations and at different time points. Abattoir records contained scarce information about the animals taken for slaughter and farmers did not keep detailed information about their animals. Thus, information about animals’ origin, age, or prior treatment was often unavailable. We did not collect weather information, which could have helped to evaluate the seasonal and yearly differences we found.

In conclusion, the in vitro susceptibility to TCBZ in field populations of *F. hepatica* parasites was associated with geographic, seasonal, and morphometric variations. Further phenotypic and genotypic characterization of flukes at the extremes of the TCBZ susceptibility spectrum could aid in understanding the epidemiology and identifying markers of resistance.

## 4. Materials and Methods

### 4.1. Parasite Procurement 

We prospectively studied the in vitro TCBZ.SO susceptibility among adult *F. hepatica* parasites collected from naturally infected bovids slaughtered in the Cusco (elevation 11152 ft) and Cajamarca (elevation 9022 ft) regions of the Peruvian highlands (Figure 2). Trained field personnel retrieved condemned bovine livers in slaughterhouses and dissected the intra- and extrahepatic biliary tree on the spot to collect adult parasites measuring ≥ 10 mm long. We collected up to 30–50 flukes per liver and batched them together throughout the study. The parasites were placed in warm RPMI + Anti medium, prepared with RPMI 1640 (Sigma-Aldrich, St. Louis, MO, USA) supplemented with 100 units/mL of penicillin, 100 µg/mL of streptomycin, and 0.25 µg/mL of amphotericin B (Antibiotic-Antimycotic, Gibco, Carlsbad, CA, USA), for transportation to our laboratories located 30 min to 3 h away. If it was known that an animal was treated for fascioliasis in the last 6 months, the liver was not used for fluke collection. Upon arrival at the laboratory, flukes were washed 5 times with normal saline solution at 37 °C. Twenty-four fully motile parasites (see below for motility scoring) were incubated at 37 °C/5% CO_2_. Parasites were individually placed and incubated in 12-well plates containing 3 mL of RPMI + Anti medium supplemented with 5% fetal bovine serum (Biowest, Riverside, MO, USA), and the medium was changed at 6 h and then every 12 h for 48 h. (Figure 3). 

### 4.2. Definition of Parasite Viability and Experimental Conditions

We used a visual motility score to assess the viability of *F. hepatica* in vitro [25,52]. Parasites were assigned a score of 3+ if they exhibited active undulating movement, 2+ if the movement was slow or decreased but still easily perceptible by the naked eye, 1+ if the movement was only perceptible by observation under 20× magnification using an inverted microscope, and 0+ if movement was not detected after observation under 20× magnification using an inverted microscope for two minutes [25,52]. 

The optimization of the experiments was performed in our Cusco site with flukes collected from three selected local slaughterhouses. We used a stepwise approach to define the TCBZ.SO exposure conditions that would identify the wildtype parasites with the highest or lowest susceptibility to the drug. TCBZ.SO (Vetranal, Sigma-Aldrich, St. Louis, MO, USA) was first dissolved in dimethyl-sulfoxide (DMSO) (Sigma-Aldrich, St. Louis, MO, USA) and then diluted in RPMI + Anti medium at 37 °C to prepare the exposure medium. Only 12 fully motile flukes from each batch were selected randomly for exposure experiments after the initial 48-h incubation period to decrease the chance of sampling clone mates. Each selected fluke was incubated at 37 °C/5% CO_2_ in individual wells of 12-well plates containing 3 mL of TCBZ.SO exposure medium. To optimize the conditions to identify susceptible and resistant flukes, the starting TCBZ.SO concentrations were 7.5 μg/mL (16.6 μM) and 15 μg/mL (33.2 μM), respectively, and the exposure time was set at 12 h [25,53,54,55]. Each plate included four control flukes incubated under the same conditions but exposed to RPMI + Anti supplemented only with DMSO at the same concentration used in the TCBZ.SO solutions. After exposure, the medium was replaced with RPMI + Anti medium supplemented with 5% FBS and exchanged every 24 h thereafter until the end of the observation period. The plate was considered invalid if >1 control fluke had a motility score of 1+ or 0+ before the end of the observation period. We titrated the drug concentrations and exposure times until we consistently identified 20% of the flukes exposed to “susceptible” conditions with motility scores of 0+ or 1+ within the observation period and 80% of the flukes exposed to “resistant” conditions with motility scores of 0+ or 1+ within the observation period. 

### 4.3. Susceptibility Testing of Wildtype F. hepatica 

The final exposure conditions to identify susceptible flukes were TCBZ.SO at 15 μg/mL (33.2 μM) for 12 h with 48 h of observation. The final exposure conditions to identify resistant flukes were TCBZ. SO at 15 μg/mL (33.2 μM), exchanged every 12 h for 24 h with 72 h of observation. Each fluke was incubated at 37 °C/5% CO_2_ in a single well of 12-well plates containing 3 mL of one of the TCBZ.SO exposure medium or control medium for the specified time and then the solutions were exchanged for RPMI + Anti supplemented with 5% FBS every 24 h for the duration of the observation period. Flukes exposed to susceptible conditions that developed motility scores of 0+ or 1+ within the time of exposure (12 h) or the time of observation (48 h) were considered susceptible and those with motility scores 2+ or 3+ undetermined. Flukes exposed to resistant conditions with motility scores of 2+ and 3+ after 24 h of exposure and 72 h of observation were considered resistant and those with motility scores of 0+ or 1+ were considered undetermined. We collected flukes during the years 2020 and 2021 from slaughterhouses in Cusco and Cajamarca. The three largest and most accessible slaughterhouses in each region were selected for weekly *F. hepatica* collections. Information on the batch, collection location, month, year, parasite length, and previous TCBZ treatment was collected when available. Fluke numbers varied per month and per site due to COVID-19 disruptions.

### 4.4. Statistical Analysis

Data were analyzed using the Statistical Package for the Social Sciences software version 25.0 (SPSS, Inc., Chicago, IL, USA). Proportions and means ± standard deviations were calculated to describe the distribution of categorical and continuous variables as appropriate. Chi-square and Student T-tests were used to compare the distribution of the categorical and continuous variables, respectively. A *p*-value < 0.05 was used to define statistical significance. 

## Figures and Tables

**Figure 1 pathogens-11-00625-f001:**
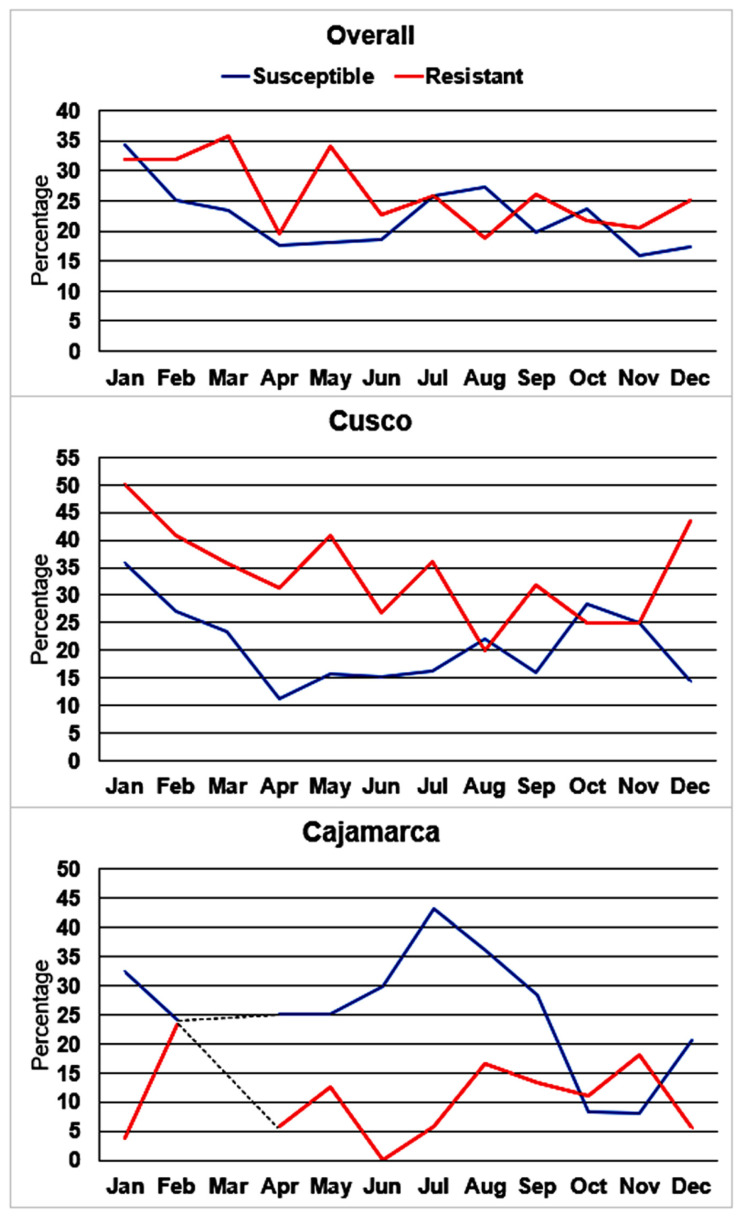
Overall monthly variation in the distribution of *F. hepatica* specimens classified as susceptible or resistant combining years 2020 and 2021. Significant (*p* < 0.05) monthly variations in the distribution of susceptible or resistant parasites were evident in the overall experiments and at each site. (dotted line = no data).

**Figure 2 pathogens-11-00625-f002:**
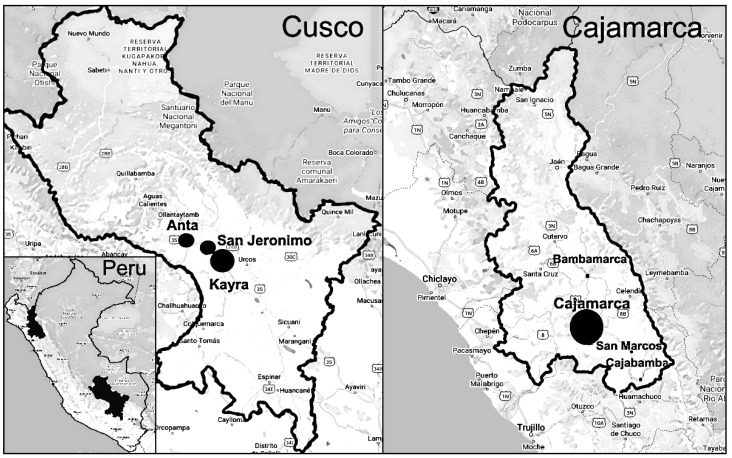
Map of the Cusco and Cajamarca regions of Peru showing the location of the slaughterhouses (the size of the circles is proportional to the number of parasites collected at the site).

**Figure 3 pathogens-11-00625-f003:**
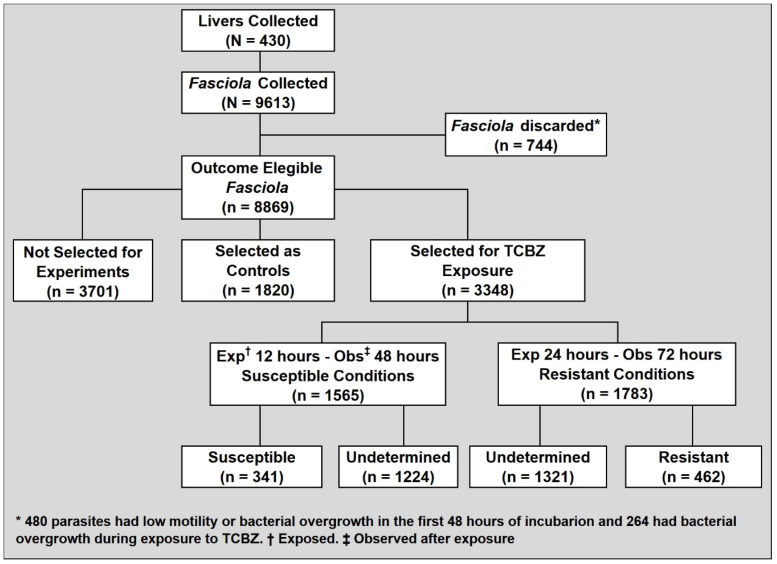
Flow diagram of the *F. hepatica* specimens collected and the selection procedures for testing.

**Table 1 pathogens-11-00625-t001:** Characteristics of the *F. hepatica* specimens according to TCBZ.SO exposure conditions.

		Total (%)	Exposed 12 h (%)	Exposed 24 h (%)
		N = 3348	N = 1565	N = 1783
Site	Cusco	2074 (61.9)	966 (61.7)	1108 (62.1)
	Cajamarca	1274 (38.1)	599 (38.3)	675 (37.9)
Slaughterhouse	Kayra	983 (29.4)	455 (29.1)	528 (29.6)
	Anta	528 (15.8)	252 (16.1)	276 (15.5)
	San Jeronimo	563 (16.8)	259 (16.5)	304 (17)
	Cajamarca	1170 (34.9)	547 (35)	623 (34.9)
	San Marcos	16 (0.5)	8 (0.5)	8 (0.4)
	Bambamarca	72 (2.2)	36 (2.3)	36 (2)
	Cajabamba	16 (0.5)	8 (0.5)	8 (0.4)
Month *	January	228 (6.8)	96 (6.1)	132 (7.4)
	February	415 (12.4)	167 (10.7)	248 (13.9)
	March	152 (4.5)	68 (4.3)	84 (4.7)
	April	296 (8.8)	148 (9.5)	148 (8.3)
	May	200 (6)	100 (6.4)	100 (5.6)
	June	216 (6.5)	92 (5.9)	124 (7)
	July	294 (8.8)	143 (9.1)	151 (8.5)
	August	191 (5.7)	95 (6.1)	96 (5.4)
	September	384 (11.5)	192 (12.3)	192 (10.8)
	October	312 (9.3)	156 (10)	156 (8.7)
	November	368 (11)	164 (10.5)	204 (11.4)
	December	292 (8.7)	144 (9.2)	148 (8.3)
Year *	2020	624 (18.6)	204 (13)	420 (22.9)
	2021	2724 (81.4)	1361 (87)	1363 (76.4)

** p* < 0.05.

**Table 2 pathogens-11-00625-t002:** Comparison of characteristics between *F. hepatica* specimens classified as susceptible or resistant to TCBZ.SO and those classified as undetermined.

Variable	Category	Susceptible N (%)	Undetermined N (%)	*p*	Resistant N (%)	Undetermined N (%)	*p*
Site	Cusco	197 (20.4)	769 (79.6)	0.089	375 (33.8)	733 (66.2)	<0.001
	Cajamarca	144 (24)	455 (76)		87 (12.9)	588 (87.1)	
Year	2020	44 (21.6)	160 (78.4)	0.935	149 (35.5)	271 (64.5)	<0.001
	2021	297 (21.8)	1064 (78.2)		313 (23)	1050 (77)	
Month	January	33 (34.4)	63 (65.6)	0.003	42 (31.8)	90 (68.2)	0.008
	February	42 (25.1)	125 (74.9)		79 (31.9)	169 (68.1)	
	March	16 (23.5)	52 (76.5)		30 (35.7)	54 (64.3)	
	April	26 (17.6)	122 (82.4)		29 (19.6)	119 (80.4)	
	May	18 (18)	82 (82)		34 (34)	66 (66)	
	June	17 (18.5)	75 (81.5)		28 (22.6)	96 (77.4)	
	July	37 (25.9)	106 (74.1)		39 (25.8)	112 (74.2)	
	August	26 (27.4)	69 (72.6)		18 (18.8)	78 (81.3)	
	September	38 (19.8)	154 (80.2)		50 (26)	142 (74)	
	October	37 (23.7)	119 (76.3)		34 (21.8)	122 (78.2)	
	November	26 (15.9)	138 (84.1)		42 (20.6)	162 (79.4)	
	December	25 (17.4)	119 (82.6)		37 (25)	111 (75)	
Slaughterhouse	Kayra	84 (18.5)	371 (81.5)	0.024	183 (34.7)	345 (65.3)	<0.001
	Anta	52 (20.6)	200 (79.4)		98 (35.5)	178 (64.5)	
	San Jeronimo	61 (23.6)	198 (76.4)		94 (30.9)	210 (69.1)	
	Cajamarca	141 (25.8)	406 (74.2)		82 (13.2)	541 (86.8)	
	San Marcos	3 (37.5)	5 (62.5)		3 (37.5)	5 (62.5)	
	Bambamarca	0 (0)	36 (100)		0 (0)	36 (100)	
	Cajabamba	0 (0)	8 (100)		2 (25)	6 (75)	

## Data Availability

The data presented in this study are openly available in the Harvard Dataverse repository at https://doi.org/10.7910/DVN/H7KTCV (accessed on 21 May 2022).

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
