# Peer review of "The Differences in the Susceptibility Patterns to Triclabendazole Sulfoxide in Field Isolates of Fasciola hepatica Are Associated with Geographic, Seasonal, and Morphometric Variations"

_pathogens, 2022, doi:10.3390/pathogens11060625_

Round 1

Reviewer 1 Report

The manuscript by Fernandez-Baca et al., characterised Fasciola hepatica triclabendazole susceptibility, by collecting fluke from naturally infected livestock at abattoirs in two regions of Peru, and cultured them with triclabendazole sulfoxide in vitro. Factors such as location, monthly time period and morphology were statistically investigated to identify any differences between fluke susceptibility and the described variables. Comments/suggestions have been made below, which I hope are constructive and will result in improving the manuscript further.

  1. In the introduction, further information on TCBZ’s proposed mode of action, what causes resistance, TCBZ metabolites produced and why TCBZ–SO was the metabolite chosen to use in experiments, would improve the manuscript.
  2. The introduction explains the importance of hepatica control in the study area. Further description of the worldwide impact of fascioliasis, would support the importance of the study.
  3. Fluke motility scoring can be subjective, how was this measurement standardised?
  4. What were the weather conditions during sample collection in 2020 and 2021?
  5. Ethical considerations of fluke collection are not mentioned in the methodology.
  6. It is not written clearly that one fluke was put in one well of a 12-well plate in 3 ml media, if this is correct, please could this be clarified in the manuscript.
  7. Why were 3,701 fluke not selected for further experiments, shown in the flow chart?
  8. Are the drug concentrations and drug exposure times used in experiments representative of what flukes would be exposed to during dosing in the field environment?
  9. Please clarify in more detail how fluke were classified as susceptible, resistant or indeterminate in the methods.
  10. In Tables 1 and 2, as fluke length has different measurement units compared to the rest of the variables in the table, this information should only be described in the text.
  11. Were any differences between fluke TCBZ susceptibility identified when fluke were collected from cattle and sheep?
  12. Should the year fluke were collected be analysed together if there was a significant difference in resistant fluke between year of collection? Would this finding be due to weather conditions?
  13. Inclusion of a map showing the amount of fluke that were susceptible and resistant at each abattoir, possibly using pie charts, would provide a geographical representation of the results.

Reviewer 2 Report

The manuscript by Fernandez-Baca and colleagues describes the analysis of the susceptibility/resistance to triclabendazole (TCBZ) within field populations of Fasciola hepatica in Peru. Their analysis was based on in vitro assays using motility scores to determine the susceptibility profile to TCBZ. The manuscript is well written, however, I question the validity of the study that is only based on motility scores – see below further comments regarding this. The authors mention that future phenotype and genotype analysis is to be carried out on these populations, but some of that data would add to this study, particularly understanding the parasite population dynamics.

  1. Line 53 and throughout the manuscript – the authors use Fasciola to describe the parasites. Are the authors referring to Fasciola spp. here or just Fasciola hepatica? Revise accordingly.
  2. Line 91-95 – the authors state that a different number of parasites were exposed to susceptible/resistant conditions, respectively. What was the reason for this difference? Also throughout the manuscript and specifically in the methods section the authors need to be clearer as to what these specific conditions are?
  3. Table 1 – Although the authors mention the 12h versus 24h exposure in the methods, this is not explained/expanded upon within the results/discussion.
  4. Lines 169-171 – The authors state that parasites were initially selected based on size, specifically over 10mm but that parasites were recovered across a varied size range which may be attributed to drug resistance. How did the authors confirm that these smaller flukes were adult parasites, rather than juvenile parasites if based on size alone?
  5. Line 213 – depending on the recovery sites for the parasites, there was a wide range in the time it took to transport the parasites back to the lab – 30min to 3 hours. Did the authors check that no obvious differences in viability could be observed between these parasites? Was this factored into the statistical analysis?
  6. Section 4.2 – Why did the authors only used a visual motility score to determine the susceptibility to TCBZ? Did the authors check the drug effects on the tegumental surface of these parasites to see if there were any obvious effects as to why the motility would be affected?
  7. Line 240-241 – To avoid selecting possible clone mates from the same animal for their analysis the authors randomly selected a small number of parasites for their in vitro assays. Although the author state that further phenotype and genotype analysis is to be completed on these populations, why did the authors not perform these assays using more parasites per animal, which could be analysed after the in vitro assays to determine if clone mates were present. It would have been interesting to investigate the dynamics between the number of parasite isolates and number of clone mates per animal and the profile of drug susceptibility in these parasites.

Reviewer 3 Report

Title: The Differences in the Susceptibility Patterns to Triclabendazole Sulfoxide in Field Isolates of Fasciola hepatica are Associated with Geographic, Seasonal, and Morphometric Variations

Authors characterized a large sample of adult Fasciola with diverging triclabendazole susceptibility in parasites from naturally infected slaughtered livestock by exposing them to triclabendazole sulfoxide in vitro to determine susceptibility and using a motility score to determine parasite’s viability. The manuscript is interesting and, again, uses a very high number of samples/dataset. Authors show valuable and interesting data. The potential impact on infectious diseases control and monitoring is high and the manuscript warrants robustness. I do have a few questions/comments that could help improve it.

Lines 82 onwards, first sentence of “Results”, I believe authors sort of repeated or summarized the goals/methods of the study “determine the distribution of the susceptibility patterns to triclabendazole sulfox-82 ide (TCBZ-SO) in field isolates of F. hepatica in the highlands of Peru, we collected 9,613 F. 83 hepatica parasites from 430 condemned bovid livers.” I would suggest avoiding this repetition.

Please revise acronyms and their definition when first used in the text

Please insert the time frame of sampling on materials and methods

Interesting (and apparently robust) visual motility score but not familiarized with it. Was it used before? Can you cite?

How many independent researchers have evaluated each parasite and what was the procedure if discordant results were present

Why the choice of chi-square and no fishers for the differences appraisal?

Round 2

Reviewer 2 Report

The authors have suitably addressed the comments raised by the previous review. The manuscript is now suitable for publication.